# Transport Membrane Condenser Heat Exchangers to Break the Water-Energy Nexus—A Critical Review

**DOI:** 10.3390/membranes11010012

**Published:** 2020-12-24

**Authors:** Jeong F. Kim, Enrico Drioli

**Affiliations:** 1Department of Energy and Chemical Engineering, Incheon National University (INU), Incheon 22012, South Korea; 2Innovation Center for Chemical Engineering, Incheon National University (INU), Incheon 22012, South Korea; 3Institute on Membrane Technology of the Italian National Research Council (CNR-ITM), 87036 Rende, Italy

**Keywords:** transport membrane condensers, membrane heat exchangers, flue gas dehydration, power plant water consumption, carbon capture process, process intensification

## Abstract

Under the notion of water-energy nexus, the unsustainable use of water in power plants has been largely accepted in silence. Moreover, the evaporated water from power plants acts as a primary nucleation source of particulate matter (PM), rendering significant air pollution and adverse health issues. With the emergence of membrane-based dehydration processes such as vapor permeation membrane, membrane condenser, and transport membrane condenser, it is now possible to capture and recycle the evaporated water. Particularly, the concept of transport membrane condensers (TMCs), also known as membrane heat exchangers, has attracted a lot of attention among the membrane community. A TMC combines the advantages of heat exchangers and membranes, and it offers a unique tool to control the transfer of both mass and energy. In this review, recent progress on TMC technology was critically assessed. The effects of TMC process parameters and membrane properties on the dehydration efficiencies were analyzed. The peculiar concept of capillary condensation and its impact on TMC performance were also discussed. The main conclusion of this review was that TMC technology, although promising, will only be competitive when the recovered water quality is high and/or the recovered energy has some energetic value (water temperature above 50 °C).

## 1. Introduction: Challenges in Power Plants

The escalating concerns over the anthropogenic impacts on the environment have led to the international COP21 Paris agreement to combat global warming [1]. The key challenge is to lower the emission of common greenhouse gases such as CO_2_, methane, and water vapor. Particularly, the major fraction of CO_2_ emission comes from the combustion of fossil fuels. As summarized in Figure 1, about 35% of all fossil fuels are used by power plants to generate electricity, followed by transportation vehicles (25%). On average, only about 39% of spent fossil fuel energy is converted to useful power, and 61% of the exothermic energy is lost as low-grade waste heat.

It is well-known that fossil-fuel-based thermoelectric power plants all over the world are major contributors to CO_2_ emissions with a high thirst for freshwater sources. Power plants also consume a surprising fraction of available water, up to 41% in the United States in 2015 [2,3], closely followed by the agriculture industry (37%). In fact, thermoelectric power plants have been the largest water consumption sector since 1965 [4]. Such thirst for water in power plants has largely been accepted in silence, as electricity supply must be maintained to meet the demand. The water drawn from power plants is mostly used for cooling and condensing the low-pressure steam leaving the gas turbine. The situation is similar in the EU where 45% of available water is used for power generation.

In South Korea, power plants employ approximately 5% of freshwater sources. Since Korea is a peninsula, power plants can be fortunately located near the sea, and most of the cooling water is saline water. South Korea’s annual cooling water consumption in 2013 was 16.7 billion m^3^, and 99% was seawater. Even so, some power plants built far from the coastlines had to shut down in 2015 due to severe droughts and water shortages. In addition, the impact of such a high withdrawal rate on the aquatic environments is still under intense debate. The cooling water leaves the power plant at an elevated temperature, disrupting the natural equilibrium. There are also reports that claim water intake structures at power plants can kill billions of fish annually [4].

Apart from the issues with unsustainable CO_2_ emission and water consumption, there is another challenge with air pollution and smog. Power plants’ flue gases and evaporated cooling water are emitted from the stack and cooling towers, respectively, at near-saturated conditions. These water vapors act as a nucleating site to combine with SO_x_ and NO_x_ derivatives to form particulate matter (PM) that pollutes the air and create unpleasant smog in the sky. In fact, 72% of all PM is indirectly generated this way [5]. Such smog is a significant issue in China and South Korea, forcing people to wear PM-protective masks.

Ironically, water is required to produce electricity, and electricity is required to produce freshwater. This notion is commonly referred to as the water-energy nexus, and recently this concept has been expanded into the water-energy-environment nexus. It is, therefore, necessary to improve the fuel and water efficiency in power plants to alleviate the water-energy-environment nexus.

A typical thermoelectric power plant operates as schematized in Figure 2. The power plant operation to generate electricity can be roughly split into three streams. The first stream is the circulating boiler water (extremely high quality), the second stream is the flue gas stream from the combustion of fossil fuels, and the third stream is the aforementioned cooling water stream.

Fossil fuels are combusted in a combustion chamber with air, which evaporates the boiler feed water. The high-pressure steam turns the turbine to produce electricity, and the low-pressure steam outlet is recondensed into liquid water then recycled back into the boiler. And it is this recondensation unit operation that requires an excessive amount of cooling water.

There are three different types of water-cooling systems: once-through, closed-cycle, and dry cooling [4]. The once-through system takes in river or seawater to condense the steam, then directly re-emits out to the environment. This type is simple and convenient but consumes the most water at a rate of 4.5 × 10^4^ m^3^·h^−1^ for a typical 500 MW plant. Thus, this type is no longer used in new power plants. On the other hand, the closed-cycle system continuously circulates the cooling water within the plant (shown in Figure 2). The cooling water temperature is maintained by evaporating a controlled portion in cooling towers shown in Figure 2. The dry cooling system does not employ cooling water and utilizes only an air stream to condense the steam. Expectedly, this system requires a significant air stream with a large footprint and is relatively inefficient. Therefore, it is only used in remote areas far from water sources.

The burnt fuels convert to flue gas that contains SO_x_ and NO_x_ derivatives. Such gases are absorbed in the flue gas desulfurization (FGD) unit using a lime slurry. The flue gas leaving the FGD unit is generally reheated to prevent unwanted corrosion and leaves the stack saturated at a temperature of 50–80 °C. A typical composition of flue gas is summarized in Table 1. It is important to note that the flue gas is a fruitful source of water and energy. For instance, lignite coal consists of 40% moisture, and a 600 MW power plant running on lignite coal exhausts 16 vol% water flue gas, which converts to a water vapor flow rate of 4.3 × 10^6^ kg·h^−1^ [7]. In addition, the amount of water vapor loss from power plants in China was estimated to be 1 billion tons per year with 2540 billion MJ, corresponding to 100 million tons of coal [8].

Therefore, to cope with the intensifying anthropogenic impact of power plants on the environment, researchers have been working to reduce CO_2_ emission and water consumption rate. Mainly, the use of heat exchangers to condense the water vapor stream has been considered [7,9]. However, the quality of recovered water is very low, and the equipment is prone to corrosion with high maintenance costs. Furthermore, liquid absorbents and solid adsorbents have been partly implemented [10,11], but the energy required for sorbent regeneration renders the installed unit uneconomical.

Among the considered separation methods, membrane technology can offer a promising solution to capture the evaporated water and waste heat from the power plant flue gases. In this review, different types of membrane-based water capture processes were assessed and compared. Particularly, the use of transport membrane condensers (TMCs) as a membrane heat exchanger (M-HEX) to capture both water and energy is the key topic of this paper.

## 2. Membrane-Based Flue Gas Dehydration: Three Different Technologies

The impact of membrane technology now is reaching beyond the well-known conventional applications of water treatment [13] and gas separation [14,15] into new exciting areas of biopharmaceutical purification [16] and power generation [17]. With better membrane materials and process innovations, the boundary of membrane technology will continue to expand wider.

A membrane has been actively applied for flue gas dehydration. As illustrated in Figure 3, the first obvious option was to employ dense vapor permeation membranes to selectively permeate water vapor over other non-condensable gases. Vapor permeation membranes have already been applied for drying natural gas [18] and compressed air [19]. Water vapor generally has much higher solubility and diffusivity compared to other gases, and hence its permeability and selectivity are in the order of 10^7^ (H_2_O/N_2_) and 10^5^ barrer, respectively [20].

Nymeijier and Wessling et al. [12] were the first pioneers to take the vapor permeation membrane technology for the flue gas dehydration process to the pilot-scale in 2008. They tested two different types of hollow fiber membranes, PEBAX 1074 and SPEEK, on real flue gas conditions in a 450 MW coal-fired power plant.

Although the permselectivity of vapor separation membranes is impressive, it cannot be fully utilized in the real processes because the saturation vapor pressure of water at the flue gas outlet condition (50–80 °C) is only about 0.1 bar to 0.5 bar. Thus, thermodynamically, only a small differential pressure can be applied, and the membrane performance is completely dependent on the permeate vacuum capacity or the sweep rate. Another very important point that must be considered for dehydration membrane is the inevitable water condensation within the membrane pores. Unlike dense films, the permeating water will condense in the support layer of hollow fiber membranes due to capillary condensation, potentially lowering the membrane permeability [21].

The test data with artificial flue gas showed a steady water capture rate of 0.6–1.0 kg·m^−2^·h^−1^ for 150 hr, and the data with real flue gas showed a water removal rate of 0.2–0.5 kg·m^−2^·h^−1^ for 5300 hr [12]. Unfortunately, the permeate water quality was not sufficiently high to be re-used directly in the steam cycle (boiler-feed water). Based on the obtained results, the required membrane area for a 450 MW coal-power plant was calculated to be 10^7^ m^2^. This pioneering work certainly validated that water capture with vapor permeation membrane is feasible, but requires further process optimization.

A membrane condenser (MC) is another innovative process to capture water vapor from flue gas, illustrated in Figure 3b. It was first proposed and developed by researchers at ITM-CNR in Italy [22,23,24,25] and was recently reviewed by Brunetti, Macedonio, Barbieri, and Drioli [26]. The key idea is to employ hydrophobic polymer membranes (pore size in the range of 0.1 and 0.2 μm) to initiate heterogeneous condensation of water vapors at the surface. In addition, by cooling the inlet stream before the membrane unit, the condensed moisture in the inlet stream can be effectively “captured” by the hydrophobic membranes. The same concept can be expanded to control various contaminants in the gaseous streams such as siloxanes, halides, VOCs, etc.

In this process, the driving force for surface condensation is not the temperature difference but the heterogeneous nucleation at the membrane surface (Figure 4). After the initial transient state, the membrane surface temperature equilibrates to the feed stream temperature due to the high latent heat of condensation. Therefore, surface chemistry must be carefully studied and tailored to maximize the condensation rate [27]. The advantage of using a hydrophobic membrane is to prevent liquid water from penetrating through the pores and allowing it to simply roll down by gravity.

Researchers at ITM-CNR have extensively investigated different aspects of membrane condensers, including the effects of the temperature gradient, feed flow rate, relative humidity, membrane materials, and process configurations on water recovery and water quality [22,24,26].

The third possible membrane-based dehydration process is the transport membrane condenser (TMC), illustrated in Figure 3c. The concept of the TMC was independently developed and proposed by Wang and Liss et al. [28,29] from the Gas Technology Institute (GTI) in the United States. The primary difference between an MC and TMC is that, in a TMC, the condensed water permeates through the membrane, and the dehumidified gaseous stream does not go through the membrane.

The authors employed a nanoporous ceramic membrane to recover both water and latent heat from the flue gas stream. The work, funded by the DOE from 2000, has reported industrial demonstration of the TMC process in gas-fired boilers, and the GTI successfully commercialized the technology. The reported performance of the TMC is very impressive. The commercialized TMC process operated for over 15,000 h, achieved a 19% reduction in greenhouse gas emission, and a 20% reduction in boiler feed water [28]. The technology has been licensed to the Cannon Boiler Work (CBW) company and a TMC product for small-scale boilers is commercially available with the trade name Ultramizer [30].

Since the first successful launch of TMC technology by the GTI, there has been many follow-up works by other researchers around the globe. In this review, we compiled recent progress and updates in this exciting new topic of membrane technology.

## 3. Transport Membrane Condensers

The TMC process can be considered as a hybrid of a heat exchanger and membrane condensers. The primary application of the TMC now is to recover both water and energy from flue gas simultaneously. A typical TMC process employs hydrophilic ceramic membranes with high thermal conductivity to condense water vapors within the membrane pores. Compared to other types of dehydration methods described above, the TMC process requires a cooling water stream to extract the thermal energy and to maintain the necessary temperature gradient to induce condensation.

The concept of TMC is very interesting and it must be approached from several angles. Firstly, the heat transfer efficiency of the TMC needs to be compared to a conventional HEX. Secondly, the peculiar concept of capillary condensation must be understood and utilized. Thirdly, the process parameters and membrane structures need to be optimized toward the TMC process.

### 3.1. Heat Transfer Efficiency of the TMC

First, the advantage of the TMC over conventional impermeable heat exchangers (HEXs) should be compared. Researchers have investigated the use of an impermeable HEX mainly to recover the latent heat [31]. Compared to other HEX configurations, modeling a HEX in high moisture conditions is very complex due to the formation of heat and mass boundary layers in the presence of non-condensable gases. A theoretical foundation was laid out by Levy and Jeong et al. [7,9,32] which showed that heat transfer efficiency of a condensation HEX can be 3.5-fold higher compared to the simple convection HEX.

Bao et al. [33] built upon the previous models to incorporate a TMC-based membrane HEX. Fascinatingly, the results show that a TMC HEX can exhibit a 50%–80% higher Nusselt number compared to the impermeable HEX at the same condensation conditions. The primary reason for such enhancement is the permeation of condensate water from the HEX surface. In the TMC, the condensate forms within the membrane pores and permeates through the membrane. In contrast, a condensate film layer forms on the surface of an impermeable HEX, appending significant thermal resistance and lowering the heat transfer efficiency. Their work certainly opened an exciting topic not only for flue gas dehydration but potentially many other applications with condensable vapors.

In the TMC, the inlet cooling water gets heated by the condensing vapor, absorbing their latent and sensible heat. However, with respect to the thermal energy recovery, the usefulness of the heated water is yet questionable. In order for the outlet water stream to be energetically useful and valuable, it needs to be heated up to at least 50 °C or higher. If the outlet water stream temperature is below 50 °C, it has very low energetic value and may not justify the use of the TMC over other available technologies. This is one of the most important aspects of the TMC and is discussed further below.

Encouragingly, the numerical results by Soleimanikutanaei and Lin et al. [34,35,36] show that the water exit temperature can be elevated up to 57 °C. However, most other works on the TMC have only focused on water extraction, and the value of an outlet water stream was not assessed in detail. In South Korea, for example, the latent heat recovered from the boiler steam vapor (outlet cooling water stream shown in Figure 2) circulates through a small city to heat up the households before discharge.

### 3.2. Capillary Condensation

In TMC technology, it is very important to understand the peculiar concept of capillary condensation and its impact on TMC performance. The saturation vapor pressure of water is primarily a function of temperature and it can be estimated using the Antoine equation. At the flue gas outlet temperatures of 50–80 °C, the water saturation vapor pressure is about 0.1–0.5 bar. However, thermodynamically, the saturation pressure can decrease within constrained curved spaces, such as in membrane pores [37,38]. It can be theoretically estimated using the Kelvin equation, as calculated in Figure 5.
(1)$$\frac{\rho \mathrm{RT}}{M}\ln\left(\frac{P_{c}}{P_{o}}\right)=\;\frac{-2\sigma \;cos\left(\theta \right)}{r_{p}}$$
where ρ is the density of the condensate, R is the universal gas constant, T is the absolute temperature, *M* is the molecular weight of the condensable compound, *P_c_* is the capillary condensation pressure, *P_o_* is the vapor saturation pressure at a planar interface, *σ* is the interfacial tension, *θ* is the contact angle, and *r_p_* is the membrane pore radius.

The water saturation pressure at 100 °C in an open atmosphere is 1 bar. However, within the membrane pores, as shown in Figure 5a, the saturation pressure decreases. Particularly, a sharp decrease can be predicted below 20 nm pore sizes. Membrane researchers have investigated this phenomenon since the 1970s on porous Vycor glass material [39] and have shown that mass transfer through a membrane via a capillary condensation mechanism can be significantly faster than vapor permeation. Asaeda et al. [40,41] exploited this interesting phenomenon to separate azeotropic mixtures (alcohol/water) with high selectivity. In theory, higher selectivity can be achieved as liquid condensate within the pores can prevent the transport of other gases.

Wang et al. [28] proposed to exploit this phenomenon on the TMC process. The authors have shown that water transfer via a capillary condensation mechanism can be 5-fold faster compared to that of the Knudsen diffusion. In addition, the separation ratio (with respect to non-condensable gases) improved by a factor of 100.

Based on this report, many follow-up works on the TMC claim to exploit the capillary condensation phenomenon, yet the experiments were not designed to clearly show its effect. Chen et al. [8] specifically designed an experiment to show that capillary condensation indeed takes place and that it improves dehydration efficiency. The authors in this paper increased the cooling water temperature higher than the feed gas dew point and showed that there is still detectable water flux. More work needs to be carried out to clearly prove the capillary condensation effect on TMC performance.

Notably, from the engineering perspective, it is not strictly necessary to induce capillary condensation as long as the temperature gradient between the feed gas and cooling water is maintained. The temperature gradient will spontaneously condense the water vapors onto the membrane surface. Therefore, many reported works simply employed membranes with pore sizes much larger than 20 nm (up to 1 μm) to test the TMC process. In fact, the report by Li et al. [42] specifically confirmed that surface condensation occurs more than capillary condensation for 1 μm ceramic membranes.

It should be stressed that the concept of capillary condensation only becomes important when the outlet water quality needs to be high. For instance, Wang et al. [28,29] claimed that the extracted water quality was high enough to be used as the boiler feed water, which must be ultrapure distilled water quality. This result is surprising given that the permeated water quality from the vapor permeation membrane did not meet the boiler feed water requirement [12]. Further investigation is necessary on this front as well.

### 3.3. TMC Data and Parametric Study

It can be seen that the TMC is relatively a complex process that requires careful tuning of operation parameters. Since the first pioneering reports of Wang et al., [28,29] many researchers have investigated the use of the TMC in lab-scale and in pilot-scale.

Relevant literature on TMC-based dehydration, to the best of the authors’ knowledge, is compiled and summarized in Table 2. Several trends can be deduced from the reported literature on the TMC. First, although scattered, the data show similar trends with respect to the TMC performance. The key operational parameters, such as flue gas temperature, cooling water temperature, flue gas flow rate, cooling water flow rate, and flue gas humidity, have been thoroughly investigated with respect to water flux and heat transfer efficiency.

Qualitative trends have been well summarized by Gao et al. [43]. Most of the obtained trends are intuitive and logical. For example, water recovery efficiency is proportional to the temperature difference between the flue gas and cooling water. In addition, the efficiency of heat recovery mostly improved with water recovery in tandem. However, unfortunately, it is yet difficult to compare the literature data, and new dimensionless numbers should be developed for the TMC process.

Secondly, TMC performance does not seem to depend on the membrane pore size. It can be seen in Table 2 that TMC water flux lies in the range between 1 and 40 kg·m^2^·hr, and there is no correlation with membrane pore size. This is an important trend, as it suggests that the current TMC process may be completely condensation-limited, not permeation-limited.

The work by Kim et al. [6] showed that water flux has a linear relationship with the inlet water vapor flow rate (shown in Figure 6a). Regardless of the operation conditions, the obtained water flux was linearly proportional to the inlet water vapor flow rate. Such a relationship indirectly indicates that water flux is not permeation-limited, as all the water condensed on the membrane surface is transferred. Note that the observed water flux was much lower than the maximum possible flux of the membranes (about 90 kg·m^−2^·h^−1^ at ΔP of 0.03 bar). Li et al. [44] also obtained similar results, where most resistance is in the condensation step, not in the membrane permeance.

This result potentially hints that employing nanoporous membranes (<20 nm) is preferred from the water quality perspectives, but such ceramic membranes are certainly more expensive. The effect of membrane pore size on selectivity is discussed below.

On the other hand, the dehumidification efficiency plateaus at around 85% (Figure 6b) due to the lack of thermal driving force above this point. A similar maximum plateau was observed in other works [8,35,45].

Thirdly, pilot-scale studies have also been carried out with up to 50,000 membrane bundles with different configurations [42,44,46]. Many systematic experimental and numerical studies have shown that staggered configuration of membrane bundles have a higher Nusselt number and hence better TMC performance.

### 3.4. TMC Materials (Ceramic vs. Polymeric)

Compared to vapor separation and conventional membrane condenser processes, all of the employed TMC membranes are inorganic ceramic membranes, as shown in Table 2. The main reason is the high thermal conductivity of ceramic materials, which is necessary to efficiently extract the heat energy from the condensing surface into the cooling waterside. The thermal conductivity of porous ceramic membranes can be estimated with different methods, and it varies widely from 1.4 to 21.84 W·m^−1^·K^−1^ [44]. In comparison, typical polymer thermal conductivity is in the range of 0.1–0.5 W·m^−1^·K^−1^ [63]. Thus, from the heat transfer perspective, the use of ceramic membranes is highly preferred.

However, ceramic membranes are seldom applied in real membrane processes due to their high cost and difficulties in handling (brittleness). In addition, a significant amount of membrane, up to 10^5^ m^2^ for a 500 MW plant, would be required to handle the voluminous flue gas flowrate. A ceramic membrane module generally has a much lower module area/volume ratio (m^2^/m^3^) than polymeric membrane modules.

On the other hand, polymeric membranes may offer a promising alternative. Polymeric membranes can be more cost-competitive, and they are mechanically more flexible and resistant. In addition, a much higher area/volume ratio is possible with a hollow fiber module. However, the thermal conductivity of porous polymeric material is relatively low. Kim et al. [6] calculated and compared the surface temperature and radial temperature profile of ceramic and polymeric membranes, as shown in Figure 7. It can be seen that the steady-state surface temperature of polymeric membranes can be 5 to 10 °C higher than that of ceramic membranes. Such analysis indicates that the TMC is certainly feasible with polymeric membranes, albeit less efficient.

Nevertheless, we expect that polymeric membranes will become a cost-effective option for the TMC in near future. An interesting study can be carried out in the future to increase the thermal conductivity of polymeric membranes by incorporating fillers, yielding mixed matrix membranes with high thermal conductivity. For instance, carbonous fillers such as graphene nanoplates possess a thermal conductivity as high as 5000 W·m^−1^·K^−1^ [63]. In addition, extended fins could be implemented onto the membrane outer surface to improve the heat transfer rate.

### 3.5. Water Dehydration Performance Comparison and Water Purity

Figure 8 summarizes the water flux between the two membrane-based dehydration processes. Based on the simple water flux comparison, the TMC may be 10- to 100-fold more effective in the flue gas dehydration process. However, more importantly, two other factors must be compared simultaneously: the quality of the recovered water and the usefulness of the recovered energy.

First, the quality of the recovered water must be discussed in detail. When the TMC was first introduced by the GTI company, Wang et al. [28,29] claimed that captured water quality from the TMC is boiler-feed quality. However, the reported data have not been reproduced by other works. The quality of boiler feed water must be extremely high [64,65] as it is ultrapure RO water. Even the water captured via dense vapor separation membranes [12] did not pass the quality threshold. Thus, there is not enough data yet to confirm that TMC water can be used for boiler feed water. Nevertheless, the quality of captured water is sufficiently high to be used for other uses such as cooling water and cleaning water.

The quality of recovered water depends on its ability to reject NO_X_ and SO_X_ contaminants. Several works investigated SO_x_ flux in the TMC process [6,48,56]. Although flue gas passes through an FGD unit, it still contains ppm-level NO_x_ and SO_x_ contaminants. SO_2_ reacts with O_2_ to form $SO_{3}^{2-}$, which immediately reacts with H_2_O vapor to form H_2_SO_4_ and exists in equilibrium as $HSO_{4}^{-}$. Kim et al. [6] reported that TMC membranes show unusually high rejection toward SO_x_ derivatives despite their small size (<1 nm) compared to the membrane pore size (20 nm). SO_x_ selectivity over water vapor reached above 100, which is much higher than the simple Knudsen selectivity of 1.8 (square root of molecular weight ratio). The exact mechanism of this selectivity is unknown, but it could be attributed to higher preferential condensation of water vapor within the membrane pores. The work by Cheng et al. [48] reported similar trends.

On the other hand, Gao et al. [56] investigated the desulfurization efficiency of the TMC using composite ceramic membranes. The objective of this work was to facilitate the permeation of SO_x_ derivative into the cooling waterside. Clearly, there is yet no consensus on SO_x_ permeation in the membrane community. However, the higher the quality of permeated water, the higher the value; hence, it is desirable to yield high SO_x_/water selectivity. In addition, more works on NO_x_ selectivity need to be investigated, as it is generally more troublesome than SO_x_ derivatives in power plants.

Another very important issue in TMC technology is the usefulness of the recovered energy. In the TMC, flue gas thermal energy is extracted into the cooling water. In order for this energy to have some energetic value, the outlet temperature should be at least 50 to 60 °C. Although researchers of the GTI company reported the outlet temperatures can reach up to 57 °C [33,34,35,36], most lab-scale results in Table 2 maintained constant cooling water temperature and did not investigate the outlet temperature. In fact, some works reported the temperature increases only a few degrees. It should be noted that the outlet temperature is mostly a process parameter that can be controlled, not the membrane parameter. The process and operation parameters need to be controlled to yield outlet temperatures above 50 °C. More experimental and numerical analysis is required on this front as well.

From an engineering perspective, a water stream with a temperature around 30 °C has almost no value; hence, we cannot claim to have recovered any energy from flue gas. But a stream with a temperature above 50 °C may have some energetic value, although not highly valuable. One potential usage of 50 °C water is to apply as membrane distillation (MD) feed water; such a process could be interesting, even feasible, if seawater is used as the TMC feed water.

## 4. Other Applications for TMC Technology

Most TMC works now are focused on the flue gas dehydration process. Recently, researchers proposed an interesting process to implement the TMC (also known as membrane heat exchangers, M-HEX) in the carbon capture process [49,52,53,54,62]. The two most promising carbon capture processes now are amine scrubbing and gas separation membranes. More specifically, amine scrubbing is currently more cost-competitive, but it consumes significant thermal energy in the stripping unit to regenerate the absorbent. The researchers proposed to integrate a TMC unit above the stripping unit to recover the latent heat of the water/CO_2_ outlet stream. This innovative idea is very promising and needs more detailed investigation.

Similarly, the TMC can be applicable to processes that emit condensable vapors, such as industries that require cooling towers. However, again, TMC technology will only be competitive if the recovered water quality is high and/or the recovered energy has energetic value (temperature above 50 °C).

## 5. Conclusions

In this review, we compiled and compared three different membrane-based dehydration processes, namely, vapor permeation membranes, membrane condensers, and transport membrane condensers (TMCs). Among the discussed processes, TMC technology exhibits a unique advantage that both water and energy can be recovered from flue gases. Many works have been reported to exploit this advantage, but the current literature seems scattered without a clear objective. In summary, the future research direction on TMC technology should be focused on the following objectives:-The effect of pore size in the TMC is still vague from the capillary condensation perspectives, and the recovered water quality needs to be investigated in more detail. The SO_X_/water selectivity as a function of membrane pore size is still an open question.-Most TMC materials now are inorganic, but more research should be carried out with cost-competitive polymeric materials. More specifically, the thermal conductivity of polymeric materials needs to be improved, possibly by incorporating fillers.-It is yet not possible to cross-compare TMC literature data objectively. There is yet no figure of merit nor dimensionless parameter in this emerging field. Thus, a new dimensionless parameter should be developed.-The energetic value of the recovered water must be assessed. In order to claim that the TMC recovered “energy,” the outlet temperature must possess useful thermal energy with >50 °C.

The most important aspect, and perhaps the main conclusion, of this review is that TMC technology will only be competitive when the recovered water quality is high enough for boiler feed water, and/or the recovered energy has energetic value, e.g., outlet water temperature above 50 °C. The current water consumption rate of power plants is certainly not sustainable, and new measures must be implemented to recover waste water and latent energy. TMC technology can be a promising option to tackle this environmental challenge and can also be applicable to CO_2_ capture processes.

## Figures and Tables

**Figure 1 membranes-11-00012-f001:**
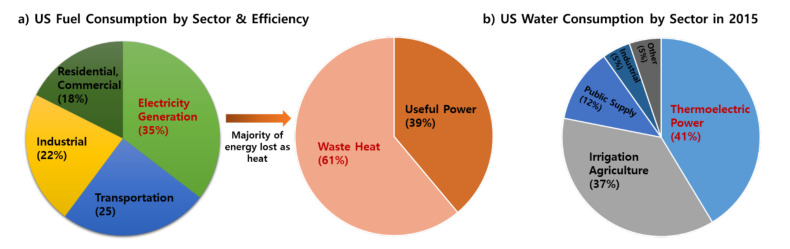
(**a**) USA fuel consumption by sector and corresponding energy efficiency, (**b**) USA water consumption by sector in 2015 (data obtained from [2,3]).

**Figure 2 membranes-11-00012-f002:**
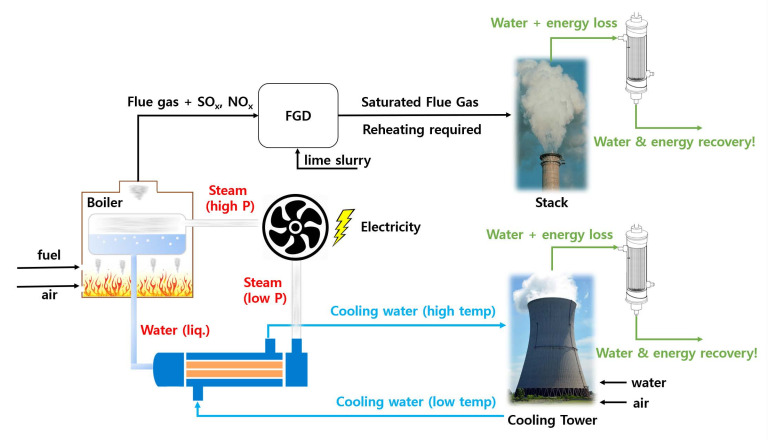
Schematized diagram of power plant operation and sources of water vapor loss. (Reprinted with permission from [6].)

**Figure 3 membranes-11-00012-f003:**
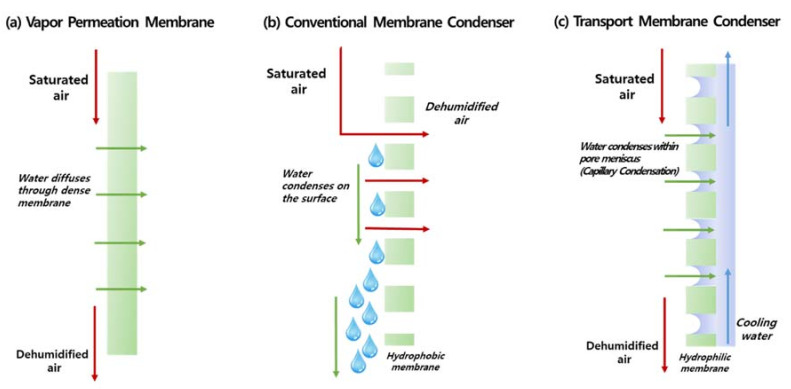
Different types of membrane-based water dehydration: (**a**) vapor permeation using a dense membrane, (**b**) conventional membrane condenser configuration using a hydrophobic microporous membrane to condense the water vapor on the surface, (**c**) transport membrane condenser using a nanoporous membrane to selectively condense water vapor within capillary pores. (Reprinted with permission from [6].)

**Figure 4 membranes-11-00012-f004:**
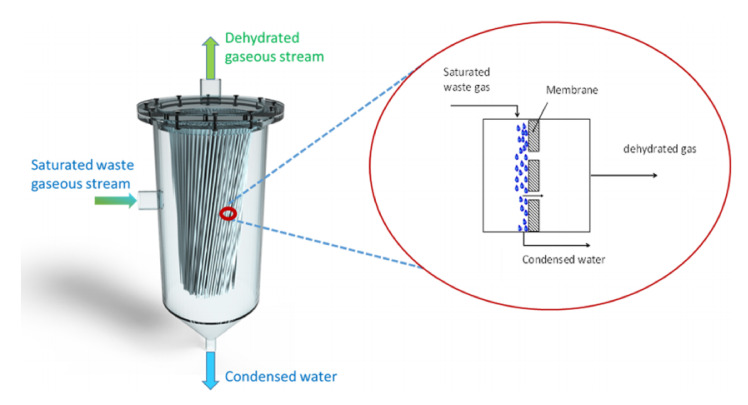
Capturing water vapor from a waste gaseous stream (Reprinted with permission from [26].)

**Figure 5 membranes-11-00012-f005:**
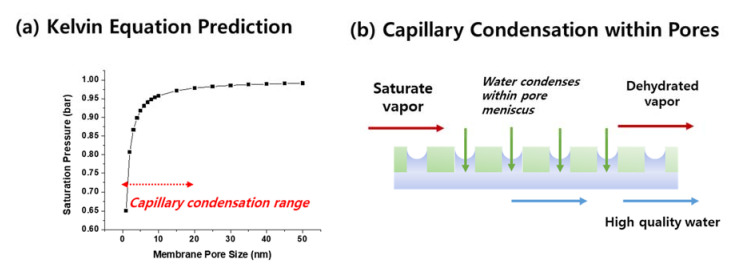
(**a**) Estimation of water saturation pressure at 100 °C as a function of membrane pore size, (**b**) schematic illustration of capillary condensation of water vapors within membrane pores.

**Figure 6 membranes-11-00012-f006:**
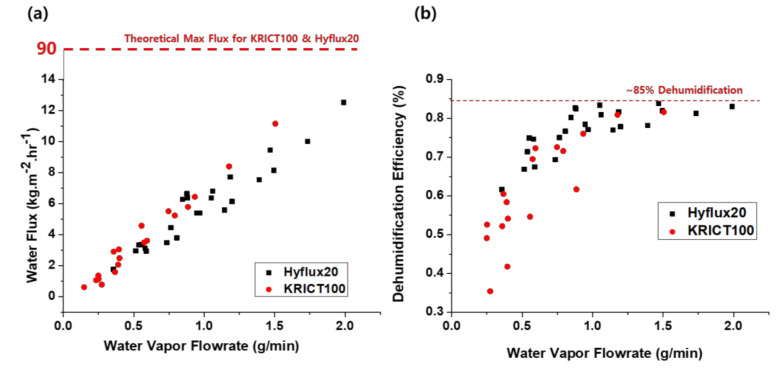
(**a**) Water flux plotted against the inlet water vapor flow rate, showing a linear relationship. (**b**) Dehumidification efficiency, performance comparison, and dehumidification efficiency (reprinted with permission from [6]).

**Figure 7 membranes-11-00012-f007:**
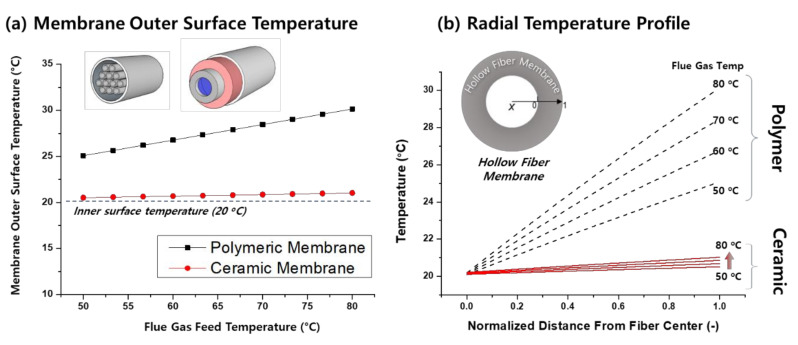
Comparison of ceramic and polymeric materials on (**a**) membrane outer surface temperature and (**b**) radial temperature profile as a function of flue gas temperature (reprinted with permission from [6]).

**Figure 8 membranes-11-00012-f008:**
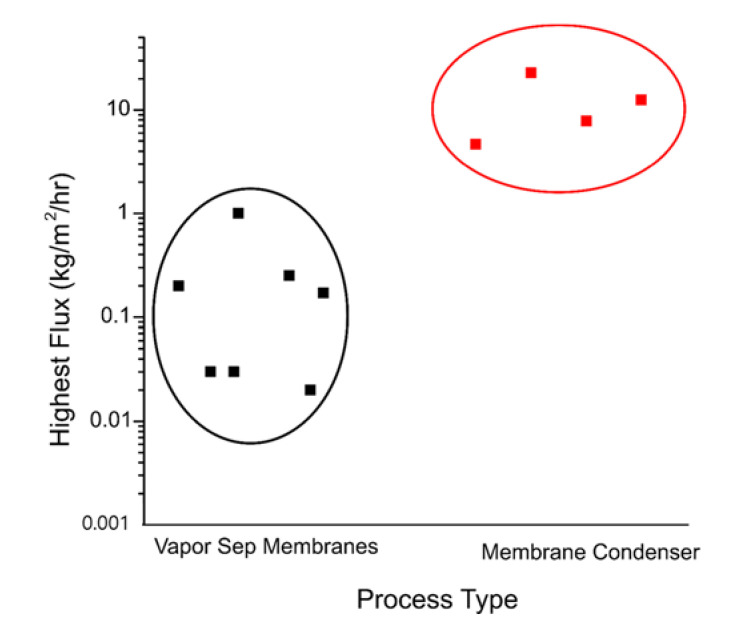
Comparison of membrane-based dehydration processes. Vapor separation membranes are polymeric and TMC membranes are ceramic.

**Table 1 membranes-11-00012-t001:** Typical flue gas composition after flue gas desulfurization (FGD) [12].

Species	Composition
H_2_O	11.2 vol%
CO_2_	13.6 vol%
N_2_	71.8 vol%
O_2_	3.4 vol%
NO_x_, SO_x_ derivatives	150–300 vppm, 50–100 vppm

**Table 2 membranes-11-00012-t002:** Summary of literature data on the TMC process.

Membrane Pore Size (Material)	Inlet Gas Temperature (°C)	Cooling Stream Temperature (°C)	Water Flux (kg/m^2^/h)	Heat Flux (MJ/m^2^/h)	Note	Year, Ref
1 μm (Ceramic)	49–53	30–36	10–22.23		Flue Gas, Pilot Scale	2021 [46]
1 μm (Ceramic)	40–60	15–35	0.5–2.93	3.6–9	Flue Gas, Lab Scale	2020 [47]
30, 50, 200 nm (Ceramic)	45–60	15–30	1.7–4.8		Flue Gas, Pilot Scale, SO_2_	2020 [48]
2, 12, 30 nm (Monolith Ceramic)	90–110	45–66	2–3	4–5	CO_2_ Capture, Lab-scale	2020 [49]
1 μm (Ceramic)	50	22	5–43.65	75–110	Flue Gas, Pilot Scale	2020 [44]
1 μm (Ceramic)	50	24–36	20–35	60–80	Flue Gas, Pilot Scale, Numerical	2020 [42]
0.4 nm (NaA zeolite sieve)	35–55	12–38	2–16		Flue Gas, Lab Scale	2020 [50]
1 μm (Ceramic)	40–50	20–25	16–22	50–70	Flue Gas, Lab Scale	2020 [51]
4 nm, 10 nm (Ceramic)	90–110	45–65	6–14	27–30	CO_2_ Capture, Lab-scale	2019 [52]
4 nm (Ceramic)	90–110	45–65	6–14	27–30	CO_2_ Capture, Lab scale	2019 [53]
4 nm Ceramic	90–110	15–60	4–6		CO_2_ Capture, Lab scale	2019 [54]
1 μm (Ceramic)	40–60	20–32	15.8		Flue Gas, Lab Scale	2019 [55]
1 μm (Ceramic)	40–60	20–32	15.77	15	Flue Gas, Lab Scale	2019 [43]
10 nm (Ceramic)	25–70	30–50	1.5–2.2		Flue Gas, Lab Scale, SO_2_	2019 [56]
6–8 nm (Ceramic)	70–80	20–43	1–7	10–15	Flue Gas, Numerical	2019 [35]
40, 90 nm (Ceramic)	50–80	20	1–13		Flue Gas, Lab Scale, SO_2_	2019 [6]
13 nm (Ceramic)	62	20	10		Flue Gas, Lab Scale,	2018 [57]
20, 30, 50, 100 nm (Ceramic)	50–70	N/A	1–3		Flue Gas, Lab Scale	2018 [58]
6–8 nm (Ceramic)	70–80	20–43	N/A		Flue Gas, Numerical	2018 [34]
N/A	50–90	10–20	10		Flue Gas, Numerical	2017 [59]
20 nm (Ceramic)	80–120	25–50	4–6	18–25	Flue Gas, Lab Scale	2017 [45]
20 nm (Ceramic)	50–70	16–65	1–15	2–15	Flue Gas, Lab Scale	2017 [8]
8–10 nm ceramic	45–85	N/A	2–15	5–45	Flue Gas, Lab Scale	2016 [60]
6–8 nm Ceramic	45–85	33	8–22	30–74	Flue Gas, Lab Scale	2015 [61]
N/A	90–110	45–115			CO_2_ Capture, Numerical	2015 [62]
6–8 nm (Ceramic)	65–95	20–45	N/A		Flue Gas, Numerical	2015 [33]
6–8 nm (Ceramic)	82	44	1–7.2		Flue Gas, Numerical	2013 [36]
6–8 nm (Ceramic)	65–85	33–55	3–7		Flue Gas, Pilot Scale	2012 [28]

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
