# Peer review of "Transport Membrane Condenser Heat Exchangers to Break the Water-Energy Nexus—A Critical Review"

_membranes, 2020, doi:10.3390/membranes11010012_

Round 1

Reviewer 1 Report

The review paper is very confusing. In the title, it mentions “transport membrane condensers” (TMC) and “membrane heat exchangers”. Theoretically, they can be the same devices. In general, the latter include the former. It is not correct to use both terms in the title. I am not sure which one is the key topic of this review. According to the abstract, it seems TMC is the focus of this review. The title is also logically wrong. “Nexus” cannot be overcome.

Apart from the title, there are too many problems. The following are some examples only:

1) Section 1. Introduction: Challenges in Power Plants and Membrane Condensers. However, there is little information on membrane condensers.

2) Why this review is needed?

3) Why membrane condensers? What are the advantages of membrane condensers over other technologies?

4) What are the advantages and disadvantages of the three membrane based dehydration technologies?

5) What is a condensing heat exchanger? Why do the authors use a different name?

6) I cannot see the logic between different sections. The whole manuscript is very confusing.

7) Many figures are not readable due to the low resolution.     

Overall, this review is in low quality and confusing.  

Author Response

Reviewer #1

The review paper is very confusing. In the title, it mentions “transport membrane condensers” (TMC) and “membrane heat exchangers”. Theoretically, they can be the same devices. In general, the latter include the former. It is not correct to use both terms in the title. I am not sure which one is the key topic of this review. According to the abstract, it seems TMC is the focus of this review. The title is also logically wrong. “Nexus” cannot be overcome.

Apart from the title, there are too many problems. The following are some examples only:

Answer: We appreciate the constructive criticisms by the reviewer on this review. We have tried to respond to all the concerns raised by the reviewer in this response letter. As for the title, we agree that both TMC and membrane heat exchangers (M-HEX) are the same concept. However, these two words are used interchangeably in the current literature. Particularly, TMC is now more widely used in recent papers. We wanted to attract the readers from both fields. We also agree with the reviewer that a “Nexus” cannot be overcome, so we changed it to “break the water-energy nexus” instead. Moreover, we tried to improve the flow of the manuscript and connect the sections more smoothly.

1) Section 1. Introduction: Challenges in Power Plants and Membrane Condensers. However, there is little information on membrane condensers.

Answer: We think the section title may have been misleading. We wanted to introduce the challenges in power plants first, then discss the membrane condenser technology. We have modified it to “Introduction : Challenges in Power Plants”

2) Why this review is needed?

Answer: We believe this review is timely, as power plants consume significant amount of available water and waste lot of latent heat. In addition, all the emitted water vapor acts as a nucleating source for particulate matters (PM) that pollute the air. In the past 10 years, many works on transport membrane condensers and membrane heat exchangers have been published to tackle this issue. However, the literature are scattered and there is no clear direction in research. Hence, we think this review can guide future research direction.

3) Why membrane condensers? What are the advantages of membrane condensers over other technologies?

Answer: Different types of water recovery technologies from power plants have been further discussed in the introduction, as copied here below.

“…Hence, to cope with the intensifying anthropogenic impact of power plants on the environment, researchers have been working to reduce CO2 emission and water consumption rate. Mainly, the use of heat exchangers to condense the water vapor stream have been considered [7,9]. However, the quality of recovered water are very low, and the equipment is prone to corrosion with high maintenance cost. Also, liquid absorbents and solid adsorbents have been partly implemented [10,11], but the energy required for sorbent regeneration renders the installed unit uneconomical. Among the considered separation methods, membrane technology can offer a promising solution to capture the evaporated water and waste heat from the power plant flue gases...

4) What are the advantages and disadvantages of the three membrane based dehydration technologies?

Answer:  Three different types of membrane-based dehydration technologies are discussed in detail in section 2, including the advantages and disadvantages. In summary, vapor permeation membranes can produce high quality water but requires excessive membrane area. Conventional membrane condenser can effectively dehydrate gas stream but the quality of water is low and the stream needs to be cooled below its dew point. On the other hand, transport membrane condensers (membrane heat exchangers) can extract both water and energy simultaneously. In addition, the recovered water, although debatable, seem to be in high quality and can be further improved.   

5) What is a condensing heat exchanger? Why do the authors use a different name?

Answer: Condensing heat exchanger are not membrane-based. They are simply metal-based heat-exchangers that operate with shell side vapor stream. The vapors condense on the outer surface of the heat exchanger to transfer latent heat of condensation to the bore side. On the other hand, membrane heat exchangers also permeate the condensed water through the pores, thus recovering both latent heat and clean water.

6) I cannot see the logic between different sections. The whole manuscript is very confusing.

Answer:  We have reorganized some of the sections and revised the introduction section to improve the flow of the manuscript with better connection between sections.

7) Many figures are not readable due to the low resolution.  Overall, this review is in low quality and confusing.  

Answer: We think the figure quality deteriorated during the PDF compilation. We will ask the Editor to provide higher quality figures.

Reviewer 2 Report

The manuscript addresses a problem of the unsustainable use of water in power plants. This review presents a valuable contribution to the comprehensive analysis of the recent progress on three membrane-based dehydration processes using, first of all, transport membrane condenser (TMC) as well as vapor permeation membrane and membrane condenser. The main conclusion has been clearly motivated, namely, “TMC technology, although promising, will only be competitive when the recovered water quality is high and/or the recovered energy has some energetic value (water temperature above 50 ºC)”.

References 38 and 39 are presented in capital letters, should be corrected.

Author Response

Reviewer #2

The manuscript addresses a problem of the unsustainable use of water in power plants. This review presents a valuable contribution to the comprehensive analysis of the recent progress on three membrane-based dehydration processes using, first of all, transport membrane condenser (TMC) as well as vapor permeation membrane and membrane condenser. The main conclusion has been clearly motivated, namely, “TMC technology, although promising, will only be competitive when the recovered water quality is high and/or the recovered energy has some energetic value (water temperature above 50 ºC)”.

References 38 and 39 are presented in capital letters, should be corrected

Answer:  We appreciate the reviewer for pointing this mistake out. We have revised it accordingly.

Reviewer 3 Report

The authors present a comprehensive review of such novel membrane separation techniques as transport membrane condensers and membrane heat exchangers. The review is well organized and delivers the appropriate list of state-of-the-art references. I would propose to accept this Manuscript after minor revision according to the following issues: 

1) Line 12, Abstract. What does it mean - (PM10 and PM2.5)? There is no explanation of it within the main text. Please explain or remove it. 

2) Lines 59-61. Please provide the references to support the statement that 72% of all PMs are indirectly generated via nucleation on the water vapors sites. 

3) Line 93. Table 1. Typical flue gas after FGD also includes carbon dioxide. Thus please provide the standard CO2 content in the flue gas. 

4) Lines 124-126. The water condensation within the membrane pores was studied as membrane wetting of porous support within thin-film composite membranes during CO2 absorption (see, for example, Scholes, C. A., Kentish, S. E., Stevens, G. W., & deMontigny, D. (2015). Comparison of thin film composite and microporous membrane contactors for CO2 absorption into monoethanolamine. International Journal of Greenhouse Gas Control, 42, 66-74.). Thus, maybe, the strong statement in the lines 124-126 might be reformulated. 

5) Lines 151-153. The supporting references should be provided. 

6) Lines 304-305. The supporting reference on the typical polymer conductivity should be provided. 

Author Response

Reviewer #3

The authors present a comprehensive review of such novel membrane separation techniques as transport membrane condensers and membrane heat exchangers. The review is well organized and delivers the appropriate list of state-of-the-art references. I would propose to accept this Manuscript after minor revision according to the following issues: 

1) Line 12, Abstract. What does it mean - (PM10 and PM2.5)? There is no explanation of it within the main text. Please explain or remove it. 

Answer:  PM10 and PM2.5 are average particle size of the particulate matters (PM) in micrometers. PM2.5 poses acute health risks, as it cannot be filtered by our lungs. In South Korea and East Asia in general, we suffer from these PMs quite severely.. But we agree with the reviewer that PM10 and PM2.5 is excessive information, so we unified it as just “PM”

2) Lines 59-61. Please provide the references to support the statement that 72% of all PMs are indirectly generated via nucleation on the water vapors sites. 

Answer: Thank you for pointing this missing reference out. This was a government report by South Korea Ministry of Environment. We have appended the appropriate reference.

3) Line 93. Table 1. Typical flue gas after FGD also includes carbon dioxide. Thus please provide the standard CO2 content in the flue gas. 

Answer:  Typical flue gas content in FGD is about 13.6 vol%, and it has been appended.

4) Lines 124-126. The water condensation within the membrane pores was studied as membrane wetting of porous support within thin-film composite membranes during CO2 absorption (see, for example, Scholes, C. A., Kentish, S. E., Stevens, G. W., & deMontigny, D. (2015). Comparison of thin film composite and microporous membrane contactors for CO2 absorption into monoethanolamine. International Journal of Greenhouse Gas Control, 42, 66-74.). Thus, maybe, the strong statement in the lines 124-126 might be reformulated. 

Answer:  We really appreciate the reviewer for pointing this out. We have carefully read the recommended reference that we were unaware of. We agree with the reviewer that the previous statements may have been too strong and have toned it down.

5) Lines 151-153. The supporting references should be provided. 

Answer:  Appropriate references have been appended

6) Lines 304-305. The supporting reference on the typical polymer conductivity should be provided. 

Answer: We have appended appropriate references. In addition, we have appended further information on possible nanomaterials (CNTs, graphenes, etc) that can be implemented to improve the thermal conductivity of membranes.

Reviewer 4 Report

It is an interesting review summarizing TMC and membrane heat exchangers. The membrane dehydration tech has been clearly summary. I suggest accepting this manuscript after minor revision.  

1.Line 74, I suggest the author descript more in detail about the water-cooling system. Explain the once-through, closed-cycle, and dry cooling.

2.About the Figure, I do suggest the author re-upload all figures with a higher resolution. I am not sure if the file converting problem or the author export problem.

Author Response

Reviewer #4

It is an interesting review summarizing TMC and membrane heat exchangers. The membrane dehydration tech has been clearly summary. I suggest accepting this manuscript after minor revision.  

1.Line 74, I suggest the author descript more in detail about the water-cooling system. Explain the once-through, closed-cycle, and dry cooling.

Answer:  We thank the reviewer for this comment. We have appended more descriptions about the different types of cooling systems.

“…There are three different types of water-cooling system: once-through, closed-cycle, and dry cooling [4]. The once-through system takes in river or seawater to condense the steam, then directly re-emitted out to the environment. This type is simple and convenient but consumes the most water at a rate of 4.5 x 104 m3·hr-1 for a typical 500 MW plant. Hence, this type is no longer used in new power plants. On the other hand, the closed-cycle system continuously circulates the cooling water within the plant (shown in Figure 2). The cooling water temperature is maintained by evaporating a controlled portion in cooling towers shown in Figure 2. The dry cooling system does not employ cooling water and utilize only air stream to condense the steam. Expectedly, this system requires significant air stream with large footprint, and is relatively inefficient. Hence it is only used in remote areas far from water sources…”

2.About the Figure, I do suggest the author re-upload all figures with a higher resolution. I am not sure if the file converting problem or the author export problem.

Answer: We think the figure quality deteriorated during the PDF compilation. We will ask the Editor to provide higher quality figures.

Round 2

Reviewer 1 Report

The previous questions have not been address.

Membrane heat exchanger and transport membrane condenser are not two parallel terms. They cannot appear in the title at the same time. Otherwise, it is repetitive.

The introduction needs to be re-written; it should focus on the membrane heat exchangers rather than power plants. Otherwise, it is out of the scope of title.

Why this review is needed?

What are the advantages and disadvantages of the three membrane based dehydration technologies?

What is a condensing heat exchanger? Why do the authors use a different name?

“2. Membrane-based flue gas dehydration: three different technologies”, “3. Transport Membrane Condensers”, “4. Other Applications for TMC Technology”, what is the logic between these sections?  

Author Response

Thank you for giving us the opportunity to revise our manuscript. We appreciate the careful reviews and constructive suggestions. After making the suggested edits, we believe that the manuscript in its current form is substantially improved. Below are the Reviewers’ comments in black with our responses in blue.

Reviewer #1 (2nd Revision)

  1. Membrane heat exchanger and transport membrane condenser are not two parallel terms. They cannot appear in the title at the same time. Otherwise, it is repetitive.

Answer: We appreciate the reviewer’s concern about the title. We have revised the title to “Transport Membrane Condenser Heat Exchangers to Break the Water-Energy Nexus”, as this term has been used by previous researchers in the same field. [Soleimanikutanei et al., Modeling and Simulation of Cross-flow Transport Membrane Condenser Heat Exchangers, International Communications in Heat and Mass Transfer, 2018, 95, 92-97]

  1. The introduction needs to be re-written; it should focus on the membrane heat exchangers rather than power plants. Otherwise, it is out of the scope of title.

Answer: The primary application of transport membrane condenser (TMC) technology is to recover water and latent heat from power plant flue gases. In fact, TMC was specifically developed to tackle this issue. We have structured the introduction section to first inform the readers on the unsustainable operation of current power plants. Then, we introduced emerging technologies, including TMC, to tackle this issue and to break the water-energy nexus. Hence, with all due of respect, we humbly disagree with the reviewer on this point and would like to maintain the current flow in the introduction.

  1. Why this review is needed?

Answer: We believe this review is necessary and timely to summarize the past 10 years work on transport membrane condensers. As discussed in the introduction section, power plants consume significant amount of available water and waste lot of latent heat. In addition, all the emitted water vapor acts as a nucleating source for particulate matters (PM) that pollute the air. In the past 10 years, many works on transport membrane condensers have been published to tackle this issue. However, the literatures are scattered and there is no clear direction in research. Hence, we think this review can guide future research direction.

  1. What are the advantages and disadvantages of the three membrane based dehydration technologies?

Answer:  Three different types of membrane-based dehydration technologies are discussed in detail in section 2, including the advantages and disadvantages. In summary, vapor permeation membranes can produce high quality water but requires excessive membrane area. Conventional membrane condenser can effectively dehydrate gas stream but the quality of water is low and the stream needs to be cooled below its dew point. On the other hand, transport membrane condensers (membrane heat exchangers) can extract both water and energy simultaneously. In addition, the recovered water, although debatable, seem to be in high quality and can be further improved.  

  1. What is a condensing heat exchanger? Why do the authors use a different name?

Answer: Condensing heat exchangers are not membrane-based. They are simply metal-based impermeable heat-exchangers that operate with the shell side vapor stream. The vapors condense on the outer surface of the heat exchanger to transfer latent heat of condensation to the bore side. On the other hand, membrane heat exchangers also permeate the condensed water through the pores, thus recovering both latent heat and clean water.

We have changed the section title of 3.1 from “Condensing Heat Exchangers” to “Heat Transfer efficiency of TMC” to avoid confusion between terms.

  1. “2. Membrane-based flue gas dehydration: three different technologies”, “3. Transport Membrane Condensers”, “4. Other Applications for TMC Technology”, what is the logic between these sections?  

Answer: In Section 1, we discussed the unsustainable operation of power plants, particularly the water and energy lost from flue gases. In section 2, we discussed possible technologies (membrane-based) to tackle this issue and to recover water from flue gases. Then, in section 3, we discussed the main technology of this review, TMC, in detail. Finally, in section 4, we discussed other potential applications of TMC, apart from power plant flue gas dehydration. More than 95% of the research on TMC are focused on power plant flue gas dehydration. However, recently, some interesting ideas have been published using TMC, and we believe more exciting applications can be developed in the future.